# Training of Primary Chicken Monocytes Results in Enhanced Pro-Inflammatory Responses

**DOI:** 10.3390/vetsci7030115

**Published:** 2020-08-19

**Authors:** Michel B. Verwoolde, Robin H. G. A. van den Biggelaar, Jürgen van Baal, Christine A. Jansen, Aart Lammers

**Affiliations:** 1Adaptation Physiology Group, Department of Animal Sciences, Wageningen University and Research, De Elst 1, 6708 WD Wageningen, The Netherlands; aart.lammers@wur.nl; 2Animal Nutrition Group, Department of Animal Sciences, Wageningen University and Research, De Elst 1, 6708 WD Wageningen, The Netherlands; jurgen.vanbaal@wur.nl; 3Department of Biomolecular Health Sciences, Division of Infectious Diseases and Immunology, Faculty of Veterinary Medicine, Utrecht University, Yalelaan 1, 3584 CL Utrecht, The Netherlands; R.H.G.A.vandenBiggelaar@uu.nl (R.H.G.A.v.d.B.); C.A.Jansen@uu.nl (C.A.J.)

**Keywords:** innate immune memory, inflammatory response, β-glucan, flow cytometry, primary chicken monocytes, macrophages

## Abstract

Beta-glucan-stimulated mammalian myeloid cells, such as macrophages, show an increased responsiveness to secondary stimulation in a nonspecific manner. This phenomenon is known as trained innate immunity and is important to prevent reinfections. Trained innate immunity seems to be an evolutionary conserved phenomenon among plants, invertebrates and mammalian species. Our study aimed to explore the training of primary chicken monocytes. We hypothesized that primary chicken monocytes, similar to their mammalian counterparts, can be trained with β-glucan resulting in increased responses of these cells to a secondary stimulus. Primary blood monocytes of white leghorn chickens were primary stimulated with β-glucan microparticulates (M-βG), lipopolysaccharide (LPS), recombinant chicken interleukin-4 (IL-4) or combinations of these components for 48 h. On day 6, the primary stimulated cells were secondary stimulated with LPS. Nitric oxide (NO) production levels were measured as an indicator of pro-inflammatory activity. In addition, the cells were analyzed by flow cytometry to characterize the population of trained cells and to investigate the expression of surface markers associated with activation. After the secondary LPS stimulation, surface expression of colony stimulating factor 1 receptor (CSF1R) and the activation markers CD40 and major histocompatibility complex class II (MHC-II) was higher on macrophages that were trained with a combination of M-βG and IL-4 compared to unstimulated cells. This increased expression was paralleled by enhanced NO production. In conclusion, this study showed that trained innate immunity can be induced in primary chicken monocytes with β-glucan, which is in line with previous experiments in mammalian species. Innate immune training may have the potential to improve health and vaccination strategies within the poultry sector.

## 1. Introduction

Vaccinations are important to control infectious diseases in poultry. Effective vaccines induce pathogen-specific protection by the formation of specific antibodies and T cells, which are part of the adaptive immune system. Moreover, pathogen-specific memory will develop due to the formation of memory T and B cells.

It has long been assumed that this immunological memory was a unique property of the adaptive immune system. However, it is now accepted that plants and invertebrates, which lack an adaptive immune system, still have the ability to develop increased protective mechanisms against pathogens after primary exposure [1,2]. This implies that memory formation is also a feature of the innate immune system.

In recent years, studies in mammalian species have shown that in vitro stimulation of monocytes and macrophages with β-glucan from yeast *Candida albicans* cell wall resulted in increased responses after a secondary unrelated stimulation [3,4,5]. Both increased surface expression of activation markers and elevated production of pro-inflammatory cytokines were observed. These enhanced responses are ascribed to epigenetic reprogramming, mediated by DNA histone modifications of the corresponding genes [3]. Furthermore, injecting mice with a low amount of heat-killed *C. albicans* resulted in an increased survival rate, and enhanced the production of tumor necrosis factor alpha (TNFα) and interleukin 6 (IL-6) by monocytes after secondary lipopolysaccharide (LPS) stimulation 7 days later [3]. Humans vaccinated with bacillus Calmette–Guérin (BCG) showed an enhanced pro-inflammatory response after a secondary stimulation with *Mycobacterium tuberculosis* or other non-related pathogens [6]. This protective effect, which was independent of T and B cells, remained for up to a year after the initial activation [7]. The mentioned findings are referred to as trained innate immunity.

Although many studies have reported evidence for trained innate immunity in mammalian species, plants and invertebrates, knowledge on trained innate immunity in avian species is limited. In the present study, we investigated the effect of stimulation with β-glucan in primary chicken monocytes by determining surface expression of the lineage marker colony stimulating factor 1 receptor (CSF1R), and the activation markers major histocompatibility complex class II (MHC-II) and CD40. Nitric oxide (NO) production was measured to observe the pro-inflammatory responses of the cells.

More knowledge on the contribution of trained innate immunity in the induction of vaccine- and feed-mediated protection in poultry may improve the effectiveness of the current vaccination and feeding strategies. This study aims to explore the training of chicken primary monocytes. We hypothesized that primary chicken monocytes can be trained with β-glucan, similar to their mammalian counterparts. To this end, we primary stimulated chicken primary monocytes with β-glucan, β-glucan + IL-4 or LPS. A secondary stimulation with LPS was subsequently given to measure the increased responsiveness of the cells by determining the expression of cell surface markers and NO production as indicators for trained innate immunity.

## 2. Materials and Methods

### 2.1. Animals and Ethical Statement

Blood for cell isolation was derived from the high feather pecking line of the 18th and 19th generations of white leghorn chickens that were divergently selected for feather pecking behavior. These chickens were part of a study on feather pecking behavior (see [8,9]). The hens were housed in 2 m^2^ pens with wood shavings on the floor under normal housing conditions matching the guidelines for laying hens. Birds received a standard rearing diet from hatching until 8 weeks of age, and another standard rearing diet from 8 until 10 weeks of age. Water and feed were provided ad libitum. Birds received vaccinations against Marek’s disease (day 0), infectious bronchitis (day 0, week 2 and 8), Newcastle disease (week 1, 4and 10) and infectious bursal disease (week 4). Blood was collected from the wing vein from ten-week-old chickens. A heparinized syringe was used to prevent the blood from coagulating. This study was approved by the Animal Welfare Committee of Wageningen University and Research in accordance with Dutch laws and regulations on the execution of animal experiments (no: AVD104002015150 and no: AVD2015357).

### 2.2. Preparation of Microparticulate β-Glucan Suspension

Beta-glucan from the *Saccharomyces cerevisiae* cell wall (Macrogard, Orffa, Werkendam, the Netherlands) consists of non-soluble macroparticles and was therefore pre-treated to obtain a microparticulate suspension. The procedure was adapted from previously described methods [10,11]. The β-glucan was suspended in sterile DNase/RNase-Free distilled water (Invitrogen UltraPure, Carlsbad, CA, USA) and shaken at room temperature using a laboratory platform rocker overnight. The next day, the suspension was diluted with NaOH to reach a final concentration of 10 mg/mL β-glucan in 0.03 M NaOH (pH 12.4). The suspension was then heated at 70 °C for 2.5 h in a water bath with regular vortex shaking. A microparticulate suspension was created using a sterile syringe and needle (BD Microlance 27 G ¾ nr 20) by drawing the suspension up and down 2 times. This treatment of β-glucan resulted in a homogeneous suspension of microparticulates (Figure A1). The suspension was aliquoted and stored at −20 °C until further use. Homogenization of the suspension was repeated every time just before the β-glucan was applied to the cells. This microparticulate β-glucan suspension is hereafter referred as ‘M-βG’.

### 2.3. Stimulation of Primary Monocytes

The collected chicken blood was diluted 1:1 in Roswell Park Memorial Institute 1640 supplemented with 25 mM HEPES (RPMI 1640) (Gibco, Life Technologies Ltd., UK). The diluted blood was overlaid onto an equal volume of ficoll-paque (Histopaque-1119, density: 1.119 g/mL, Sigma-Aldrich corporations, St. Louis, MO, USA) to separate the leukocytes by density gradient centrifugation (700× *g*, 40 min at room temperature). The interphase containing the leukocytes was collected, washed 2 times with RPMI 1640 and re-suspended in culture medium. This culture medium contains RPMI 1640 supplemented with 25 mM HEPES, Glutamax ™, 10% heat inactivated chicken serum and 50 U/mL penicillin and 50 μg/mL streptomycin (all from Gibco).

The timeline of the ex vivo innate training experiment is shown in Figure 1. Leukocytes were seeded at a concentration of 1 × 10^6^ cells per well in a 96-well flat bottom plate (CELLSTAR, Greiner Bio-One, Alphen aan den Rijn, The Netherlands) in a total volume of 100 μL per well. The cells were incubated overnight at 41 °C in 5% CO_2_ and 95% humidity. The next day, non-adherent cells were washed away with pre-warmed (41 °C) culture medium. Adherent cells from every individual chicken were stimulated in a volume of 200 μL per well with culture medium supplemented with M-βG (10 µg/mL), LPS from *E. coli* serotype O55:B5 (10 µg/mL, L2880, Sigma-Aldrich corporations, St. Louis, MO, USA), recombinant chicken IL-4 (100 ng/mL, Kingfisher Biotech Inc., Saint Paul, MN, USA) or a combination of M-βG and IL-4. As a control, cells were incubated in culture medium without additional stimuli during the stimulation period. Cells were collected for flow cytometry analysis 24 h after stimulation. From an identical experiment performed simultaneously, cell culture supernatant was collected 48 h post-stimulation to measure the production of NO. After these 48 h, all cells were washed two times with culture medium to remove the stimuli and cultured further in 200 μL culture medium per well at 41 °C in 5% CO_2_ and 95% humidity. At D6, the cells were secondary stimulated with 200 μL LPS (10 µg/mL). Cells were collected 24 h after the secondary stimulation for flow cytometry analysis. From an identical experiment performed simultaneously, cell culture supernatant was collected after 48 h for subsequent analysis of NO production.

### 2.4. Nitric Oxide (NO) Production Assay

NO production was measured 48 h after the primary and secondary stimulation (Figure 1). NO was indirectly measured by quantifying the production of the more stable nitrite (NO_2_^−^), using the Griess reaction assay as previously described [12,13]. Briefly, 50 µl culture supernatant was transferred to a new 96-well flat-bottom plate (Greiner CELLSTAR^®^) and combined with 50 µl of Griess reagent consisting of a 1:1 mixture 2% Sulphanilamide in 5% H_3_PO_4_ and 0.2% N-(1-naphthyl)ethylenediamine dihydrochloride in H_2_O. The plate was incubated for 10 min at room temperature. The NO_2_^−^ concentration was determined by measuring the optical density at 540 nm with a spectrophotometer (Thermo scientific, Multiscan™). The results were interpolated on a standard curve made by serial diluting a sodium nitrite solution (NaNO_2_) in the range from 100 μM to 0 μM.

### 2.5. Flow Cytometry

Flow cytometry was performed 24 h after the primary and secondary stimulation to phenotypically characterize the cell populations (Figure 1). Cells were washed with PBS w/o Ca^2+^ and Mg^2+^ (Gibco) and subsequently detached with 5 mM EDTA in PBS. The detached cells were transferred to a 96-well round-bottom plate (CELLSTAR, Greiner Bio-One, Alphen aan den Rijn, The Netherlands) and kept on ice. Staining and washing steps were performed in FACS buffer containing PBS w/o Ca^2+^ and Mg^2+^, supplemented with 0.5% BSA and 0.005% NaN_3_ (Sigma-Aldrich). The cells were stained with one of the following primary mouse monoclonal antibodies: anti-chicken CSF1R (clone ROS-AV170, IgG1; Bio-Rad), anti-chicken CD40 (clone LOB7/6, IgG2a; Bio-Rad, Hercules, CA, USA), or biotin-conjugated anti-chicken MHC class II (clone Ia, IgMκ, SouthernBiotech, Birmingham, AL, USA) at 4 °C in the dark for 20 min. After washing in FACS buffer, cells were incubated with the secondary antibodies: R-phycoerythrin (PE)-conjugated goat anti-mouse-IgG1 or allophycocyanin (APC)-conjugated goat-anti-mouse-IgG2a (both SouthernBiotech), together with Alexa Fluor 405-conjugated streptavidin (Invitrogen). Cells were again washed in FACS buffer and then stained with fluorescein (FITC)-conjugated mouse-anti-chicken KUL1-(IgG1) antibody (SouthernBiotech) for a period of 20 min at 4 °C protected from light. Finally, after washing the stained cells with FACS buffer, the 7-aminoactinomycin D (7-AAD; BD) was added to exclude nonviable cells. The samples were acquired on a FACSCanto^TM^ II flow cytometer (BD Biosciences, San Jose, CA, USA). Data analysis was performed using FlowJo Software v. 10.5 (TreeStar Inc, San Carlo, CA, USA).

### 2.6. Statistical Analysis

Statistical analysis was performed using Prism version 7.04 software (GraphPad Software, San Diego, CA, USA). Differences in the mean among the experimental groups of the NO assay were analyzed using a one-way ANOVA with Tukey’s multiple comparison tests. Flow cytometry data were expressed in geometric mean fluorescent intensity (gMFI) and fold change. A two-way ANOVA with Tukey’s multiple comparison test was used for statistical analysis of the gMFI data. Fold change was calculated for each group with different primary stimulation conditions after secondary stimulation with LPS or unstimulated by $\frac{gMFI_{LPS}}{gMFI_{unstimulated}}$. A repeated measures one-way ANOVA with Tukey’s multiple comparison test was used for statistical analysis of the fold change data. *p* < 0.05 was considered a significant difference. *p*-values between 0.05 and 0.1 were considered to indicate a tendency.

## 3. Results

### 3.1. In Vitro Culture Resulted in a Highly Homogeneous Macrophage Population after 7 Days of Culture

Primary monocytes were isolated from chicken blood and characterized after 24 h (Figure 2A) and 7 days (Figure 2B) of culture by flow cytometry to get more insight into the composition of the cell population. The cells were gated for viability (7-AAD^-^), high forward and side light scatter (FSC vs. SSC), indicative of macrophages [14]. The macrophages expressed chicken macrophage markers KUL01 [14] and CSF1R [15], MHC-II and low levels of co-stimulatory molecule CD40 at both timepoints. On D7, the macrophage population was highly homogeneous, comprising >90% of the total cell population.

### 3.2. Primary Stimulation with β-Glucan Microparticulates and IL-4 Enhanced NO Production after Secondary Stimulation with LPS

We investigated the pro-inflammatory responses in chicken monocytes by measuring NO production upon primary and secondary stimulation (Figure 3 and Figure 4). The cytokine IL-4 was included in the study because IL-4 highly upregulated the expression of the major receptor for beta-glucan, dectin-1, in murine macrophages [16]. As indicated in Figure 3, NO production was increased after primary stimulation with LPS compared to the unstimulated cells (LPS: 17.23 ± 3.17, UNSTIM: 0.94 ± 0.13, *p* < 0.001). We did not observe NO production after primary stimulation with M-βG (0.88 ± 0.13), IL-4 (0.45 ± 0.12) or the combination of M-βG + IL-4 (1.01 ± 0.13) compared to the unstimulated cells. Six days later, cells were secondary stimulated with LPS.

As indicated in Figure 4, this resulted in NO production which was higher in cells which were primary stimulated with M-βG + IL-4 compared to primary unstimulated cells (UNSTIM-LPS) (M-βG + IL-4: 27.70 ± 3.89, UNSTIM-LPS: 8.76 ± 2.08, *p* < 0.001). No differences were found in NO production after secondary stimulation with LPS for the cells that were primary stimulated with M-βG (9.51 ± 2.29), IL-4 (9.44 ± 1.50) or LPS (7.19 ± 1.29) compared to primary unstimulated cells (UNSTIM-LPS). Taken together, we found an increased NO production, which is indicative of a pro-inflammatory response, after secondary stimulation with LPS in cells which were primary stimulated with M-βG in combination with IL-4.

### 3.3. Primary Stimulation with β-Glucan Microparticulates and IL-4 Influenced CD40, MHC-II and CSF1R Surface Expression after Secondary Stimulation with LPS

In addition to NO production, we investigated the expression of the surface markers KUL01, CSF1R, MHC-II and CD40. KUL01 and CSF1R are two well-known myeloid markers and were primarily used in this study to phenotypically characterize the macrophages in the cell population [14,15]. Surface markers CD40 and MHC-II are associated with activation of myeloid cells [17,18,19,20,21]. After 7 days, primary activation of the cells had gone down to baseline with respect to activation markers CD40 and MHC-II, although primary stimulation with LPS resulted in a lasting increase in KUL01 expression and primary activation with beta-glucan + IL-4 resulted in a lasting decrease in CSF1R expression (Figure 5B). Secondary stimulation with LPS resulted in increased surface expression of CD40 compared to secondary unstimulated cells (Figure 5A,C). This increase was larger for macrophages primary stimulated with M-βG + IL-4 compared to macrophages primary stimulated with LPS or to primary unstimulated cells (fold change with M-βG + IL-4: 3.67 ± 0.27, fold change with LPS: 2.36 ± 0.15, *p* < 0.01; fold change with UNSTIM: 2.65 ± 0.19, *p* < 0.05). In contrast to CD40, surface expression of MHC-II and CSF1R was lower after the secondary LPS stimulation in macrophages primary stimulated with LPS or in primary unstimulated cells (Figure 5A,C). Interestingly, the lower expression of MHC-II and CSF1R was largely prevented by M-βG + IL-4 primary stimulation (MHC-II fold change with M-βG + IL-4: 0.89 ± 0.06, fold change with UNSTIM: 0.52 ± 0.05, *p* < 0.001; CSF1R fold change with M-βG + Il-4: 0.88 ± 0.10, fold change with UNSTIM: 0.61 ± 0.05, *p* < 0.05). KUL01 expression was lower in M-βG + IL-4 primary stimulated macrophages compared to primary unstimulated cells after secondary stimulation with LPS (fold change with M-βG + Il-4: 0.83 ± 0.04; fold change with UNSTIM: 0.98 ± 0.04, *p* < 0.05, Figure 5C).

### 3.4. No Evidence of Training in Chicken Bone Marrow-Derived Adherent Cells.

Trained innate immunity is not conserved to blood monocytes only (e.g., Natural killer cells) [22]. Therefore, in parallel to the training of blood-derived primary monocytes, we tried to train chicken bone marrow-derived monocytes and myeloid progenitors using the same approach. NO production was again determined as a pro-inflammatory measure for trained innate immunity. Both primary stimulation and secondary stimulation are shown (Figure A2 and Figure A3). The responses to the primary stimulations were similar to the blood-derived primary monocytes. NO production after the primary stimulation with LPS was increased compared to the unstimulated cells (LPS: 4.35 ± 0.23, UNSTIM: 2.37 ± 0.09, *p* < 0.001, Figure A2). However, the responses to the secondary stimulation with LPS did not result in enhanced NO production for M-βG + IL-4 primary stimulated cells compared to primary unstimulated cells (M-βG + IL-4: 4.12 ± 0.38, UNSTIM-LPS: 3.80 ± 0.50, Figure A3).

## 4. Discussion

The present study is to our knowledge the first study describing trained innate immunity in primary chicken monocytes. In this study, we measured NO production and analyzed the surface expression of the markers CD40 and MHC-II which are associated with monocyte activation, indicative of a pro-inflammatory response [17,18,20,21]. Primary stimulation with M-βG in combination with IL-4 resulted in an increased immune responsiveness to LPS, reflected by increased NO production and increased surface expression of CD40, MHC-II and CSF1R. Our results are in line with previous observations on trained innate immunity in mammalian species. Hence, we confirmed our hypothesis that primary chicken monocytes are trainable with β-glucan in combination with IL-4.

Trained macrophages from mice produced more NO compared to untrained cells after secondary LPS stimulation [23]. Furthermore, trained macrophages from mice and humans showed enhanced production of pro-inflammatory cytokines such as IL-1β, IL-6 and TNFα [3,24]. Indeed, within our chicken study, training with M-βG + IL-4 resulted in increased cell surface expression of CD40 and MHC-II and elevated NO production. This association with CD40 expression fits earlier observations. In mice, it has been described that an increase in CD40 ligation was found to stimulate the expression of nitric oxide synthase (iNOS) [17]. NO, as a product of iNOS activity, is an effector molecule of activated macrophages that kills microbes within macrophages through its reactivity with proteins, DNA, thiols and iron at the active site of many enzymes [25]. Both the results on expression of the cell surface markers and NO production show that primary chicken monocytes can be trained, similarly to their mammalian counterparts.

NO production of trained monocytes may lead to increased killing capacity upon phagocytosis [26]. Indeed, a study in bovine monocyte-derived macrophages showed that increased bacterial killing capacity could be induced by macrophage training upon stimulation with heat-killed *Mycobacteria bovis* in vivo [27]. In that study, this was referred to as innate immune training.

It is known that increased phagocytosis results in enhanced antigen presentation [28]. A higher level of surface markers such as CD40 and MHC-II may lead to an enhanced adaptive immune response, since both CD40 and MHC-II play an important role in antigen presentation and the subsequent activation of the adaptive immune system [17,18,29,30]. The study with bovine monocyte-derived macrophages also indicates a relationship between trained innate immunity and increased antibody levels of the adaptive immune response [27].

Interestingly, training was only observed when monocytes were trained with M-βG in combination with IL-4. With the current read out parameters, we found no significant effects of training by M-βG and IL-4 separately. Beta-glucan is a known agonist of the pattern-recognition receptor dectin-1 found on mammalian phagocytes [31]. A positive correlation has been found between stimulation with IL-4 and/or IL-13 and surface expression of dectin-1 receptor in murine macrophages within 4 h [16]. In accordance with this observation, the addition of IL-4 to our cultures may have caused an upregulation of dectin-1 receptors, making the macrophages more responsive or accessible to M-βG. Although an intensive BLAST search in the latest *Gallus gallus* genome database (GRCg6a: build GCF_000002315.6) did not result in the identification of a dectin-1 chicken homologue, a dectin-l like β-glucan receptor is likely to be present on chicken heterophils and PBMCs (peripheral blood mononuclear cells), which have been found to respond to the dectin-1-specific agonist curdlan by an oxidative burst [32]. Whether dectin-1 or other pattern-recognition receptors for β-glucan, such as Toll-like receptor 2, Toll-like receptor 6 and Complement receptor 3, play a role in chicken innate immune training by β-glucan and IL-4 has to be elucidated and awaits further studies [31,33,34,35,36]. We did not find evidence for trained innate immunity in chicken bone marrow-derived cells, which is different from previous studies in murine bone marrow cells and may be interesting for further investigation to understand this discrepancy [37].

In humans, primary stimulation with LPS resulted in a tolerogenic state of the macrophages [24]. In the current study, no tolerance was observed, since no significant decrease was observed for LPS primary stimulated cells in the surface expression of the activation markers CD40 and MHC-II nor NO production upon secondary stimulation with LPS. The fact that we did not find evidence of LPS-induced tolerance in the current study contrasts with other in vivo studies in birds [38,39]. At this moment we are not able to clarify the absence of the tolerogenic state of the LPS primary stimulated cells. However, tolerance might be dependent on age, time of stimulation and dosage of the component, but the exact mechanism behind LPS tolerance is not fully known [40,41].

In our study, the surface markers CSF1R and KUL01 were not used as markers for training but were primarily used to phenotypically characterize the macrophages in the cell population. However, primary stimulation with M-βG + IL-4 resulted in higher CSF1R and lower KUL01 expression after secondary LPS stimulation (Figure 5C). This suggests that training affects the regulation of macrophage survival, differentiation and proliferation [15]. In line with our observation, reduced KUL01 expression was also found on bone marrow-derived monocytes after LPS stimulation [42,43].

We showed trained innate immunity by using a relatively large sample size of individual chickens (*N* = 21). Within this group of chickens, we observed substantial individual variation. Understanding these individual variations can be of great value in understanding the mechanism behind trained innate immunity. A possible explanation may be small differences in genetic background between these chickens. Another factor that may influence training are the DNA modifications that determine the activity of the genes, so called epigenetic changes [44]. These changes are independent of genetic background but are influenced by external factors such as feed, environment, age and even the parents.

In conclusion, we showed training of primary chicken monocytes. More research on, for example, cytokine production, metabolic mechanisms, and epigenetic changes will be of great value to understand the mechanisms behind trained innate immunity in chickens. Innate immune training may have potential to improve disease resistance of poultry in a nonspecific manner, especially at a young age when the adaptive immune system has not yet fully developed [13,45,46,47]. Dietary additives or vaccinations based on β-glucan could potentially be applied in vivo to train innate immune cells and improve resistance to a variety of pathogens. An in vivo experimental infection, investigating different pathogenic organisms, should assess whether in vivo training has cross protective effects and whether increased pro-inflammatory responses will not damage the host. Possible interactions of enhanced innate immunity with metabolic and/or behavioral physiology should be considered [48,49].

## Figures and Tables

**Figure 1 vetsci-07-00115-f001:**
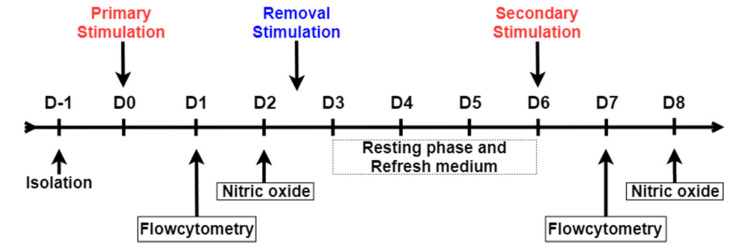
Schematic overview of the in vitro model for trained innate immunity. Adherent cells were primary stimulated with culture medium (UNSTIM), lipopolysaccharide (LPS), β-glucan microparticulates (M-βG), interleukin-4 (IL-4) and M-βG + IL-4 on day 0 (D0). On D1, cells were harvested for flow cytometry. On D2, cell culture supernatant was collected to measure the release of nitric oxide (NO), from an identical experiment performed simultaneously. Cells were subsequently washed to remove any stimuli and incubated in fresh culture medium until D6. On D6, all treatment groups were stimulated with LPS as secondary stimulation. The negative control (UNSTIM-UNSTIM) was not treated with LPS but only incubated with culture medium. Again, cells were subjected to flow cytometry analysis on D7 and NO release in the cell culture supernatant was measured on D8.

**Figure 2 vetsci-07-00115-f002:**
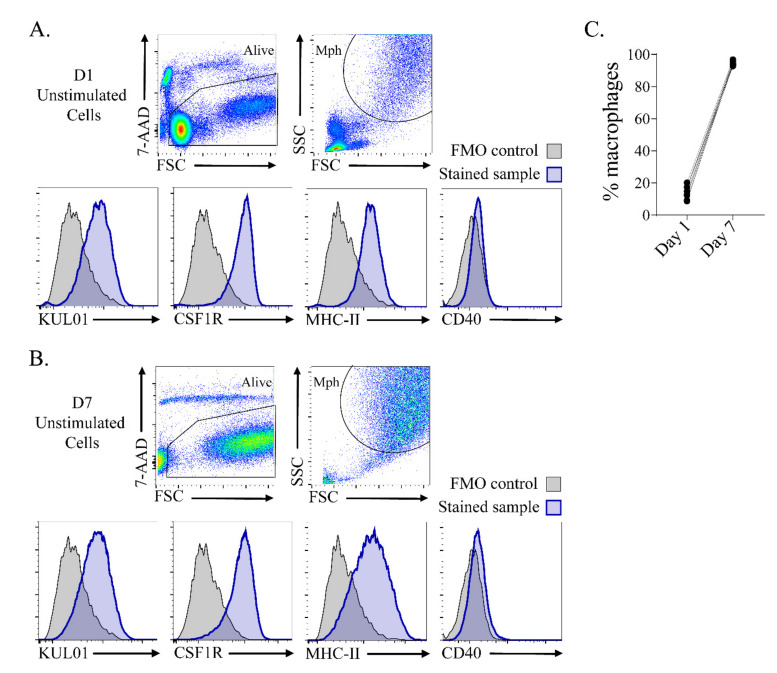
Adherent cells differentiated to KUL01^+^ CSF1R^+^ MHC-II^+^ macrophages (Mph) after 7 days of culture. (**A**) Adherent cells were characterized after 24 h of culture by flow cytometry. (**B**) Adherent unstimulated cells were characterized after 7 days of culture by flow cytometry. The cells were selected for viability (7-AAD^−^), forward and side light scatter (FSC vs. SSC), and assessed for the expression of KUL01, CSF1R, MHC-II and CD40. The histograms show expression of the macrophage markers in blue and fluorescent-minus-one (FMO) staining controls in grey. (**C**) The percentages of macrophages from day 1 and day 7.

**Figure 3 vetsci-07-00115-f003:**
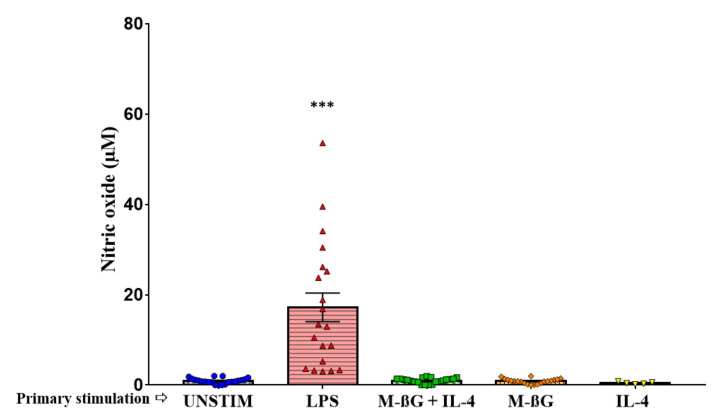
Primary stimulation with LPS resulted in enhanced nitric oxide production. Fresh isolated adherent cells were primary stimulated with culture medium (UNSTIM), LPS (10 μg/mL), M-βG (10 μg/mL), M-βG + IL-4 (10 μg/mL + 100 ng/mL) or IL-4 (100 ng/mL). LPS induced NO production (*N* = 21 chickens; N IL-4 = 5 chickens). Each bar represents mean ± SEM. *** *p* < 0.001.

**Figure 4 vetsci-07-00115-f004:**
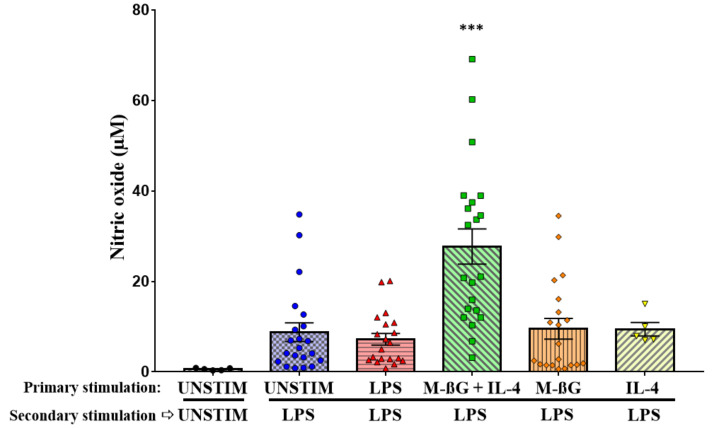
Primary stimulation with β-glucan microparticulates + IL-4 enhanced the NO production after secondary stimulation of LPS. Freshly isolated adherent cells were primary stimulated with culture medium (UNSTIM), LPS (10 μg/mL), M-βG (10 μg/mL), M-βG + IL-4 (10 μg/mL + 100 ng/mL) or IL-4 (100 ng/mL) on D0. The cells were secondary stimulated with LPS (10 μg/mL), except for the negative control (UNSTIM-UNSTIM) on D6. NO production after the secondary stimulations are shown in this figure. Only cells primary stimulated with the combination M-βG + IL-4 showed increased NO production after a secondary stimulation with LPS compared to primary unstimulated cells (*N* = 21 chickens). Each bar represents mean ± SEM. *** *p* < 0.001.

**Figure 5 vetsci-07-00115-f005:**
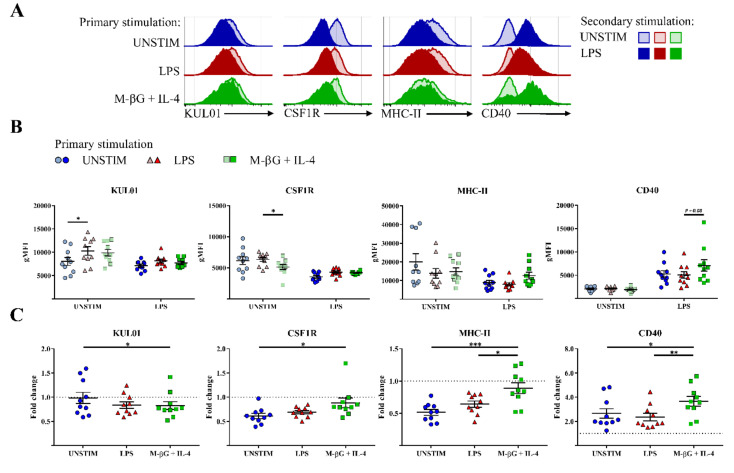
Primary stimulation with β-glucan microparticulates + IL-4 influenced the expression level of cell surface markers on macrophages after secondary stimulation with LPS. Surface expression of KUL01, CSF1R, MHC-II and CD40 was assessed after secondary stimulation with LPS (10 μg/mL). (**A**) Expression of the markers is shown by histograms for macrophages derived from one representative chicken upon secondary stimulation with LPS or secondary unstimulated cells (UNSTIM). (**B**) The effect of different primary stimulations after 7 days for secondary unstimulated cells (UNSTIM) and upon secondary stimulation with LPS. The expression of the markers KUL01, CSF1R, MHC-II and CD40 were expressed in geometric mean fluorescent intensity (gMFI). (**C**) The effect of different primary stimulations on secondary stimulated macrophages for surface expression of KUL01, CSF1R, MHC-II and CD40. The expression of the markers upon secondary stimulation with LPS was compared to secondary unstimulated cells (UNSTIM) and changes were expressed as a fold change in geometric mean fluorescent intensity (gMFI). For (**B**,**C**), each bar represents mean ± SEM (*N* = 10 chickens). * *p* < 0.05; ** *p* < 0.01; *** *p* < 0.001.

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
