# Peer review of "Training of Primary Chicken Monocytes Results in Enhanced Pro-Inflammatory Responses"

_vetsci, 2020, doi:10.3390/vetsci7030115_

Round 1
Reviewer 1 Report
The manuscript “Training of primary chicken monocytes results in enhanced pro-inflammatory responses” by Verwoolde et al. describes an in vitro model for assessing the induction of innate memory in chicken monocytes, using beta-glucan particles as priming agent.
The authors use the production of NO as marker of inflammatory cell activation, and also measure a number of surface markers. The conclusion is that beta-glucan in combination with IL-4 primes monocytes towards an enhanced response to a challenge with LPS, measured in terms of NO production and upregulation of some markers.
The study has merits in that it shows a nice model for assessing innate memory in chicken monocytes. However, some drawback need addressing.
- The characterisation of the beta-glucan microparticles should be improved. The pictures shown are of very poor quality (particle characterisation is usually done by TEM) and their size distribution should be declared. This is important, because of the several suggestions that size of particles/microorganisms may have a role in innate memory induction.
- The surface markers (KUL01, CSFR1, MHC-II, CD40) may only be associated to cell differentiation and unrelated to inflammatory activation, thus they are not expected to give any meaningful information about the functional changes induced by priming. In addition, the profiles shown in the upper part do not match with the graphs below. Knowing the difficulties in having clear-cut data out of the flow cytometry profiles, I have the feeling these data do not show anything meaningful. This part of the study should be carefully revised.
- I did not see how the authors have assessed that at day 6 the primary activation has completely gone down to baseline. I would have expected to see data showing that NO and surface markers in primed cells are at the same levels as in unprimed cells. Some of these data can be found here and there, but I would recommend to include the complete set of data in the manuscript.
- The fact that LPS tolerance is not evident is a problem that goes against the reliability of the in vitro model. I would suggest to find and include here the conditions in which this can be seen, otherwise the entire model loses credibility.
- Having NO as the only inflammation-related functional marker is frankly insufficient (the surface markers are not relevant in this direction). TNF alpha has been identified in chickens, and many reagents for measuring it are available. I would suggest to include TNF expression or protein production as a marker of monocyte activation.
- I think it is very wise that the authors claim that priming with beta-glucan enhances subsequent inflammatory responses and refrain from claiming that such training can induce protection against infections. Enhanced inflammatory reactivity may indeed be detrimental and cause significant damage to the host. Thus, in order to see whether the priming is protective, animals should be subjected to an experimental infection and their survival/death rate recorded. Otherwise, the authors should keep their discussion to speculations about the possibility of modulating innate memory in affording protection against a range of different infections, once a “protective “ memory profile has been defined upon in vivo
Reviewer 2 Report
The paper by Verwoolde et al. describes that the trained immunity is present in chicken monocytes. In particular, authors showed that the primary stimulation of monocytes with glycan particles in combination with cytokine IL-4 induces memory responses and prepares immune cells for future stimulation with such microbial agents as LPS. The fact that the addition of IL-4 was necessary to train chicken immune cells represents a very interesting observation and requires additional discussion. In addition, authors should discuss why bone-marrow derived macrophages cannot be trained as blood-isolated monocytes. Overall, the manuscript is well-written and easy to read. I highly recommend it for publication after minor corrections highlighted below.
Abstract: It makes sense to mention that trained immunity is evolutionally conserved trait in animals and represent an important (bit not yet well-understood) mechanism to prevent re-infections. Just to inform readers why the investigation of trained immunity in various species is necessary.
Line 14: more appropriate to say “myeloid cells such as macrophages “rather than just “macrophages”
Line 26: I would suggest to decipher acronyms here
Line 176: Why Figure B1 and Figure 2 are separated? Why not combine them and add a quantitative analysis and compare percentage of differentiation overtime.
Line 194: Have you considered that other receptors besides dectin 1 can be involved in initiation of trained immunity? This is interesting because training murine immune cells with betta glycan did not require IL-4.
Line 224: Why KUL01, CSF1R were analysed? Specify here.
Line 251: Why this was done?
Can it be that the expression of a receptor that is stimulated by IL-4 and recognizes glycan is absent in chicken BMDMs? What physiological consequence it may have?
Line 269: more than untrained cells?
Line 292: Citation format is incorrect
Line 316 Please discuss the absence of trained immunity in BMDMs in contrast to other cell types here and compare to what is known for mouse trained immunity
Round 2
Reviewer 1 Report
The authors have replied very well to all my comments. In my opinion, the manuscript is now ready for publication.